# Experimental Data Collection of Surface Quality Analysis of CuCrZr Specimens Manufactured with SLM Technology: Analysis of the Effects of Process Parameters

**DOI:** 10.3390/ma16010098

**Published:** 2022-12-22

**Authors:** Ilaria Caravella, Daniele Cortis, Luca Di Angelo, Donato Orlandi

**Affiliations:** 1Gran Sasso National Laboratory, National Institute for Nuclear Physics, 67100 L’Aquila, Italy; 2Department of Industrial and Information Engineering and Economics, University of L’Aquila, Piazzale Pontieri, Monteluco di Roio (AQ), 67100 L’Aquila, Italy

**Keywords:** additive manufacturing, selective laser melting, process parameters, copper, CuCrZr, surface quality, surface texture

## Abstract

Selective laser melting (SLM) is the most widely used laser powder-bed fusion (L-PBF) technology for the additive manufacturing (AM) of parts from metallic powders. The surface quality of the SLM parts is highly dependent on many factors and process parameters. These factors include the powder grain size, the layer thickness, and the building angle. This paper conducted an experimental analysis of the effects of SLM process parameters on the surface quality of CuCrZr cubic specimens. Thanks to its excellent thermal and mechanical properties, CrCrZr has become one of the most widely used materials in SLM technology. The specimens have been produced with different combinations of layer thickness, laser patterns, building angles, and scanning speed, keeping the energy density constant. The results show how different combinations of parameters affect the surface quality macroscopically (i.e., layer thickness, building angle, and scanning speed); in contrast, other parameters (i.e., laser pattern) do not seem to have any contributions. By varying these parameters within typical ranges of the AM machine used, variations in surface quality can be achieved from 10.4 µm up to 40.8 µm. These results represent an important basis for developing research activities that will further focus on implementing a mathematical/experimental model to help designers optimize the surface quality during the AM pre-processing phase.

## 1. Introduction

Selective laser melting (SLM) is the most widely used laser powder bed-fusion (L-PBF) technology for the additive manufacturing (AM) of parts from metallic powders. It is a process that melts and fuses selective powder regions layer by layer using a high-intensity laser as an energy source. Process parameters, such as laser power, scanning speed, hatch spacing, and layer thickness, must be adjusted so that a single melt vector can fuse entirely with the neighboring melt vectors and the preceding layer [1]. SLM technology offers many merits, such as high flexibility in the design that allows:product customization at an acceptable cost (due to the lack of tooling or fixtures needed);production of components with complex geometry and high spatial resolution (e.g., porous structures [2], turbine discs [3], aeronautic components [4], and lightweight cellular structures [5]);improved microstructure and properties [6].

Because the most common SLM machines use infrared laser sources with a wavelength of 1064 nm, they are primarily suitable for part fabrication by materials with low reflectivity and low thermal conductivity and are free of low boiling point volatile elements [7]. Many papers in the literature analyze the use of materials such as maraging steel [8], 316L stainless steel [9], titanium alloy [10], Inconel alloy [11], and aluminum alloy [12].

CuCrZr alloy has excellent thermal and electrical conductivity, hardness, and strength. These characteristics, together with its good castability, machinability, and excellent mechanical properties, have resulted in the use of this alloy in many high-tech applications [13]. Their utilization for manufacturing parts with SLM technology has been limited by copper’s high laser reflectivity in the infrared range (90%) and high thermal conductivity (400 W/m K at 300 K), which can easily cause melt instability and defects [14,15]. To exploit the previously mentioned advantages of SLM and CuCrZr alloys, several researchers have started to investigate the SLMed CuCrZr alloy. These efforts have been mainly focused on the effect of the process parameters and heat treatment on the microstructure and properties of the SLMed CuCrZr alloy [16]. In particular, the published research investigated the process parameters and the influence of heat treatment on increasing strength, and electrical conductivity [17,18,19,20].

All these methods ignore the analysis of a fundamental characteristic of AM objects: surface quality. This feature is critical because currently, with metal AM technologies, it is impossible to achieve surface qualities as good as those generated via CNC machining [21]. The surface quality of the SLM parts is highly dependent on many factors and process parameters. Among them, the grain size of powder particles, layer thickness, wall angle, and melt pool size are the most relevant [22]. The wall angle and layer thickness combination produce the well-known staircase effect that reduces the surface quality; this inevitable and systematic error for some values of angles and thickness makes use of the well-known roughness index $R_{a}$ inappropriate [23]. Other critical elements that cause poor surface quality for AM technologies, such as the SLM, is the required connection between the part and the building platform and supports for overhangs greater than 40° to prevent thermal deformations of the built part. In addition, the removal of the support structures is highly time-consuming and is most often carried out manually.

Janhs et al. [15] investigated the functional properties and heat treatment of SLMed CuCr1Zr alloy generated by gas atomization. Regarding the surface quality, they analyzed the surface roughness of a side and the upper surface of specimens, varying the laser power (from 100 to 370 W) and the laser scanning velocity (400, 600, and 800 mm/s). The results show that the Ra value increases with laser power, and the highest roughness was achieved with a laser power of 370 W. Furthermore, the upper surfaces exhibit a lower roughness than the side surfaces, possibly due to the multiple melting performed on the topmost powder layer. Finally, parameters leading to the formation of the better surface of samples are those that produce the worst relative optical densities. Because the typical surface roughness $R_{a}$ of the SLMed CuCrZr parts ranges from 8 to 12 µm even after process optimization and this surface quality is inadequate for meeting the requirements of many applications, the only solution is machining the most critical surfaces [16]. Bai et al. [16] proposed the analysis and comparison of the microstructure, mechanical properties, and machinability of as-built and heat-treated CuCrZr alloys fabricated with SLM. Machining experiments were conducted on the as-built and heat-treated samples to study the machinability, including cutting force, surface quality, and chip morphology. Results regarding the surface quality show that building direction and heat treatment have a significant influence on the machinability of the SLMed CuCrZr alloy. High surface quality can be achieved for surfaces parallel to the XZ plane machined after aging treatment. The conclusions of this paper are limited to applications for which post-processing on surfaces of objects manufactured with SLM technology is possible. Furthermore, an important limitation of the last two papers is neglecting a critical parameter such as the wall angle.

A helpful design support would be a mathematical/experimental model that assists the designer in global (e.g., deposition direction) and local surface-by-surface (e.g., machine parameters) choices in manufacturing an object by SLM technology. Such a model would be essential in all applications where surface quality is a necessary functional requirement and post-processing is not possible on the surfaces of manufactured objects due to their shape or accessibility. Characteristic and extreme examples are the internal surfaces of custom heat exchangers (e.g., lattice structures) and waveguides: the former requires a rough surface, and the latter requires a surface with the best surface quality possible.

This paper analyzed the effect of SLM process parameters on the surface quality of CuCrZr-manufactured cubic specimens. The main objective of the experiment is a preliminary collection of data for implementing a mathematical/experimental model to help designers optimize the surface quality during the AM pre-processing phase, considering the final purpose of the component. Particular attention was given to geometrical process parameters influencing surface quality, such as layer thickness, laser pattern, and building angle. The results show how different combinations of process parameters affect the surface quality macroscopically, and surface quality is exclusively affected by layer thickness and building angle. The laser pattern strategy does not seem to have any contribution, and the improvement in the texture is proportional to layer thickness. The tuning of the scanning speed can also be used to improve surface quality.

## 2. Materials and Methods

### 2.1. Specimen Production

CuCrZr cubic specimens (9 × 9 × 9 mm) were produced using a SISMA MySint100 PM/RM machine based on SLM technology. The machine, specific for R&D activities, is equipped with an infrared laser up to 200 W and a fixed spot diameter of 30 µm with a Gaussian spatial distribution. The process parameters, such as hatch distance (µm), scan speed (µm/s), layer thickness (µm), and laser pattern, are completely configurable by the user, and they can be changed directly onboard a machine or with classic external pre-processing software (e.g., Magics). The production of the specimens took place under an inert nitrogen (N) atmosphere with an oxygen level below 0.1% using a brass building platform. The powder material used in the research, whose chemical analysis is reported in Table 1, was supplied by Metals4Printing company.

The specimens were produced with different combinations of the following parameters:layer thickness,laser pattern,building angle.

These parameters constrained the energy density (*E*), transferred by the laser on the powder, to an optimized level (E156 J/mm${}^{3}$), as defined by Equation (Equation 1):(1)$$E=\frac{P}{v\times h\times t}$$
where *P* represents the laser power, *v* is the laser scanning speed, *h* is the hatch distance, and *t* is the layer thickness. The optimized energy density level resulted from preliminary experimentation; the reported value permitted manufactured components with a good material density without compromising the surface quality (i.e., without excessive powder overburn on the sides of the specimens), which is useful for the proposed research.

The SLM process parameters were selected using a full factorial design of experiment (DOE) with three levels (−1, 0, +1) for the variables mentioned above (i.e., *N*. $=3^{3}$ = 27 combinations). Table 2 reports the combinations and the print jobs necessary to produce the specimens.

Because the powder grain sizes range from 15 to 45 µm, the minimum layer thickness was defined at 40 µm to prevent any minimal voids in the distribution of the powder by the recoater system of the SLM machine. On the other hand, the maximum layer thickness has been defined considering the staircase effect so that no excessive approximation of side geometry is introduced. Based on the results proposed by Huxol et al. [25], according to which laser strategy affects mechanical characteristics, three different laser patterns are tested (Figure 1): linear, chess 3 × 3 mm, and strip 3 mm. Figure 1 shows the three different scanning patterns following the 90° clockwise rotation of the laser in two subsequent layers, considering the constraints by the cubic dimension of the samples.

Moreover, the building angle (θ) has been defined as the clockwise angle between the building direction (*z*-axis) and specimen surface, as reported in Figure 2. For each specimen, the blue surface (BS) and the red surface (RS) are considered with respect to the powder supply direction. As such, an evaluation of the surface quality as a function of the building angle (θ) from 0° to 150° was performed, which also analyzed the influence of the powder supplier direction. Building angles of surfaces requiring support structures were not analyzed. For example, Figure 3 shows the result of Print Job I (A# specimens).

### 2.2. Surface Quality Analysis

Although new methods for surface quality assessment have been published in the literature in recent years [26], in this paper, to make the proposed procedure repeatable and reproducible, areal methods defined by the EN ISO 25178-2 standard [27] are applied. The measurements were performed using a KEYENCE VHX-7000 Digital Microscope based on a non-contact/point autofocus probe, also known as focus variation technology, according to the EN ISO 25178-605 standard [28]. The evaluated parameters within a definition area (*A*) are reported in Table 3. A short-wavelength filter with $\lambda_{s}=10$ µm is first applied. Considering the analyzed thicknesses and angles, a long-wavelength filter with $\lambda_{s}=2.5$ mm was used. This value is greater than the characteristic dimensions of the texture surfaces considered here, so a typical texture surface with a long wavelength (the staircase effect) is not filtered, and the roughness can be considered to analyze the surface quality.

The $S_{a}$ parameter, defined as follows:(2)$$S_{a}=\frac{1}{A}\overset{}{\underset{A}{\;\int \int \;}}\left|z(x,y))\right|dxdy$$
can be directly correlated with the well-known linear surface roughness ($R_{a}$) [29].

## 3. Results and Discussion

Before proceeding with the surface quality analysis, each specimen was dimensionally controlled using a KEYENCE IM-8000 Image Measurement System. This control has been made to verify the potential presence of unexpected deformations on the specimen side surface that could have affected the surface quality analysis. The results of the non-contact measurement are reported in Table 4. Each reported value is calculated as the mean of three measures. The data show how the building uncertainties along the three dimensions of the cubic specimen are almost the same: approximately 0.1 ± 0.09 mm. The high number of samples makes the results statistically reliable, showing that all pieces are dimensionally acceptable without unexpected deformations.

In addition, before analyzing the quality of the surface of all specimens, preliminary measures were carried out on a few samples to define the correct dimension of the definition area (*A*) and to assess the repeatability of the SLM production process. According to ISO 25178-3: 2012 [27], the evaluation area *A* consists of a rectangular portion of the surface over which extraction is made. Because the form typically influences the evaluation area’s orientation, the rectangular area has to be parallel/orthogonal to the nominal geometry (e.g., cylinder axis, sides of a rectangular flat, etc.). The size of the evaluation area has the same length as the filter “nesting index”, and it is typically five times the scale of the coarsest structure of interest. Because these scales were not known a priori for the objects manufactured with the material and machine considered here, preliminary measures were performed on a few samples. The objective of this was to find the maximum size of the evaluation area for surface quality measurements unaffected by specimen edge effects; the necessity to choose the maximum size results from the requirement to measure the whole structure of interest with certainty. For this purpose, the A1, A4, and A7 specimens manufactured with the same level of variables (layer thickness = 40 µm, laser pattern = linear) but with a different building angle (θ=0° and 150°), were measured. The preliminary measures of $S_{a}$ were performed on both selected surfaces (i.e., BS and RS), progressively increasing the evaluation area sizes from a minimum of 2 × 2 mm to 9 × 9 mm (Figure 4). The results in Figure 5, each one representing the average of three measurements, highlight a significant boundary effect for a definition area (*A*) of 9 × 9 mm. By considering smaller areas, the edge effect quickly disappears as the measurements become comparable. This evidence is compatible with the edge effects of SLM technology due to the unmelted powder around the specimens, the hatch distance, and the distance of the laser pattern to the outer edge of the specimen, set at 0.08 µm during the production. Based on these considerations, the surface texture parameters ($S_{a}$, $S_{p}$, $S_{v}$, $S_{z}$, $S_{q}$, $S_{sk}$, and $S_{ku}$) were measured with a definition area (*A*) of 8 × 8 mm. With a magnification of 100×, this allowed for acquisition of 7741 × 7741 points for each texture, on which the software calculated surface quality parameters.

Three different A1 samples manufactured with the same set of parameters were analyzed to assess the repeatability of the SLM production process at varying building angles. The results for the blue (0°) and red (90°) surfaces are reported in Table 5 and Table 6. Each reported value is calculated as the mean of three measures. The data show that the objects with SLM technology manufactured with the same parameters, although they show punctual differences in the generated surfaces (see $S_{p}$, $S_{v}$, and $S_{z}$), have a relative standard deviation on $S_{a}$ of 0.5% for the blue and 1.05% for the red surface. The resulting differences are so minor that they may be caused by the randomness of the surface irregularities and measurement errors. The repeatability in the construction of the surfaces can be considered independent of the extreme values of building angles (0° and 90°).

After these preliminary evaluations, the analysis of the surface texture parameters for all specimens was performed using an evaluation area of 8 × 8 mm, and the results are summarized in Table 7 and Table 8, where the mean of the three measures are reported. All the relative standard deviations on $S_{a}$ values of the reported values vary between 0.05% to 1.68%, showing no recognizable influence on the parameters considered here. These non-zero values of the standard deviations are due to manual positioning of the acquisition area by the operator using the graphical user interface of the KEYENCE VHX-7000 Digital Microscope.

The results show how there is at most a ∼60% difference (i.e., Table 7, Specimen ID#A9 and C6) between the largest and smallest value of $S_{a}$, demonstrating how the different combination of variables affects the results macroscopically. The minimum value of $S_{a}$ is found for each group in the same combination: for BS with #9 (Table 7) and RS with #4 (Table 8). Considering the DOE analysis, the optimum conditions have been identified for the BS as a layer thickness of 40 µm, a building angle of 60°, and a linear laser pattern. For the RS, the optimum conditions were the same layer thickness and pattern strategy, but with a building angle of 120°. The analysis results are summarized in Table 9 and Table 10, where for each level of variables (−1, 0, +1), the $S_{a}$ mean value and the relative difference in identification of the main effects of variables are reported. The $S_{a}$ value is exclusively affected by layer thickness and building angle, and the laser pattern strategy does not seem to contribute anything. This experimental evidence can be explained considering that, from a geometric point of view, the surface texture is rightly influenced by the layer thickness of the building section and by its angle (i.e., the staircase effect). The laser pattern only plays a role in the selective melting of powder material without apparently changing the surface morphology. The DOE analysis returns the same indications when considering the maximum height parameter ($S_{z}$) instead of the $S_{a}$; this supports the close correlation between these two factors.

Taking the above considerations into account, the presence of any interactions between layer thickness and building angle was assessed. Figure 6, Figure 7 and Figure 8 show $S_{a}$ as a function of these two parameters, keeping the laser pattern fixed at a specific level. The plots correctly describe the influence of the layer thickness on the $S_{a}$ results for each pattern condition and how the building angle between 60° and 120° produces the best results. Moreover, the parallelism between the segments highlights no interactions between the variables, and the reduction in $S_{a}$ seems to be proportional as the layer decreases (∼5 µm).

The values obtained for surfaces with the same construction angles but oriented differently to the powder supply (30° and 150°, 60° and 120°) are then compared. Although they show punctual differences in the generated surfaces (see $S_{p}$, $S_{v}$, and $S_{z}$), the differences in terms of $S_{a}$ are so minor that they could be caused by the randomness of the surface irregularities.

The following further information can be deduced by analyzing the other surface texture parameters:the root mean square height ($S_{q}$) has the same trend as the $S_{a}$ (Figure 9a,b);the skewness ($S_{sk}$) has a mean value of about 0.5 and 0.6 for the BS and RS, respectively; this indicates that the distribution of the measures (i.e., the ordinate value z(x,y)) is almost similar to a Gaussian curve because it is not far from zero;the kurtosis ($S_{ku}$) shows that the distribution of the measures (i.e., the ordinate value z(x,y)), except for a few outliers, has a very slight peak in the mid frequencies when compared to a standard Gaussian curve, where $S_{ku}$ is zero (Table 5 and Table 6).

Starting from machine parameters corresponding to the lowest $S_{a}$ value (specimen A1, layer thickness = 40 µm, laser pattern = linear, building angle θ=0°), another experiment was carried out to evaluate the influence of the laser scanning speed. For this purpose, three specimens were manufactured with a laser scanning speed of 1.25, 2, and 4 times the value used for A1, maintaining the same level of energy density (*E*∼ 156 J/mm${}^{3}$). The results, reported in Table 11 and Table 12, again, each representing the average of three measurements, demonstrate how the $S_{a}$ can be reduced by half or even about a third with a simple adjustment of the process parameters, particularly in the last building layers, without compromising the average density of the component. This improvement is more evident for the BS than for the RS, probably due to the smaller size of the melt pool generated by the laser on the powder bed surface.

Figure 10 summarizes some of the most important results from the experimentation here presented. Figure 10a shows the interpolating surfaces of $S_{a}$ data varying the building angle and thickness for blue and red surfaces. Figure 10b shows for the vertical (blue surface) and horizontal surfaces (red surface) the trend of $S_{a}$ varying the laser scanning speed. These graphs identifying the most critical parameters affecting the surface quality of CuCrZr-manufactured parts represent the initial kernel of the mathematical/experimental model to be implemented in future works.

## 4. Conclusions

This paper presented an experimental analysis to investigate the effect of SLM process parameters on the surface quality of CuCrZr-manufactured cubic specimens. This experimentation can be considered as a preliminary collection of data for implementation of a mathematical/experimental model. That model should help the designer during the AM pre-processing phase to optimize the surface quality, considering the final purpose of the component. The reported experimentation focused on the influence of the following:layer thickness,laser pattern,building angle,laser scanning speed.

These constrained the energy density (*E*), transferred by the laser on the powder, to an optimized level (*E* ∼ 156 J/mm${}^{3}$). Furthermore, the specifically designed specimen shape enabled the investigation of the influence of the power supplier direction. Surface quality was evaluated using areal methods defined by the EN ISO 25178-2 standard [27]. Preliminary experimentation was conducted on a few samples to define the dimension of the evaluation area (A) and to assess the repeatability of the SLM production process. The results highlight that, for specimens with nominal sizes of 9 × 9 mm, the maximum evaluation area sizes not affected by the edge effect is 8 × 8 mm. This fact shows a significant edge effect due to unmelted powder on the edge of the specimens. Concerning the SLM repeatability, better results are obtained in building surfaces facing the powder supplier (i.e., building angle between 60° and 120°). Through extensive experimentation, it was demonstrated that different combinations of process parameters affect the surface quality macroscopically, and the surface area is exclusively affected by layer thickness and building angle. The laser pattern strategy does not seem to have any contributions, and the reduction in $S_{a}$ is proportional to layer thickness. Finally, it has been verified that increasing the laser scanning speed can reduce the $S_{a}$ by half or about a third. This makes it possible to improve surface quality, especially in the last layers, without compromising the average density of the component. Further investigations will expand experimental data by analyzing the other building angles to complete the maps of Figure 10. In addition, future efforts will be addressed to investigate other methods for characterizing surface quality, such as multi-scale methods [26].

## Figures and Tables

**Figure 1 materials-16-00098-f001:**
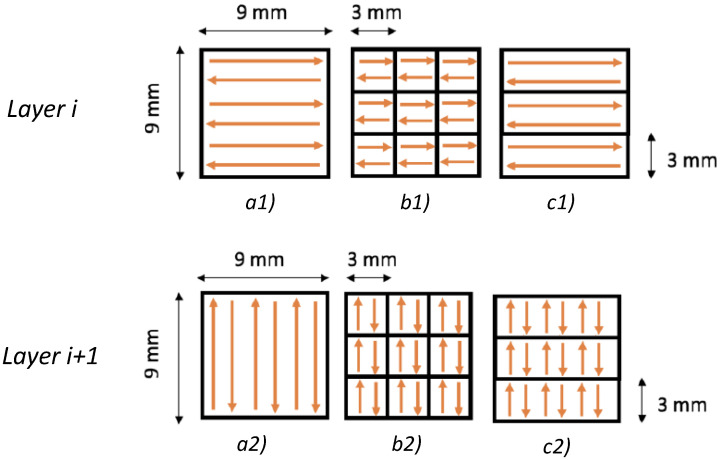
Laser pattern characteristics for two subsequent layers: (**a1**) Linear, (**b1**) chess 3 × 3 mm, and (**c1**) strip 3 mm at layer i-th; (**a2**) Linear, (**b2**) chess 3 × 3 mm, and (**c2**) strip 3 mm at layer i+1-th.

**Figure 2 materials-16-00098-f002:**
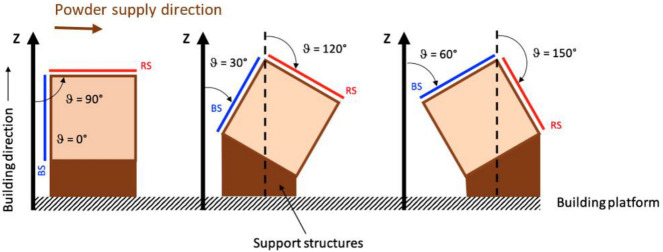
Building angle (θ) definition and analyzed surfaces: blue surface (BS) and red surface (RS).

**Figure 3 materials-16-00098-f003:**
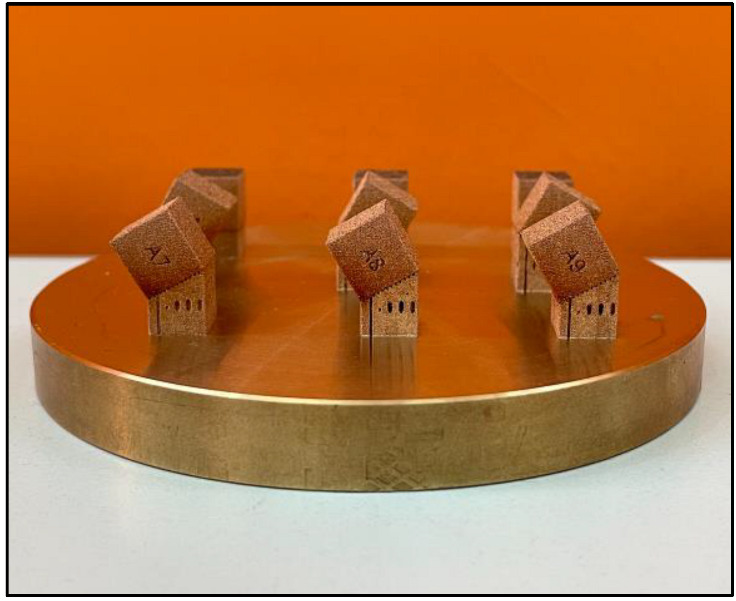
Print Job I: A# specimens’ production results on the building platform.

**Figure 4 materials-16-00098-f004:**
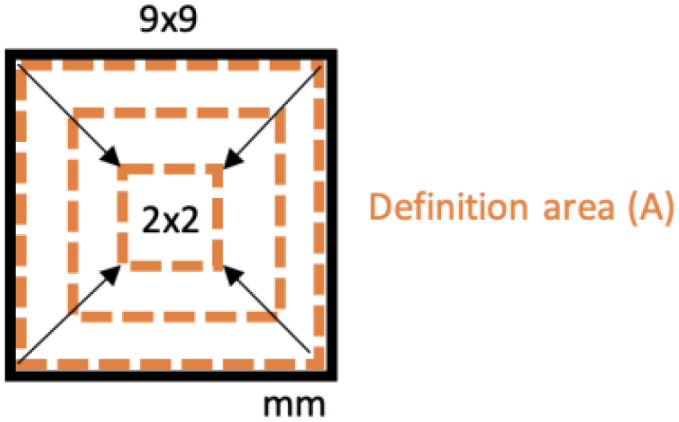
Definition area (*A*) evaluation: influence of boundary effects.

**Figure 5 materials-16-00098-f005:**
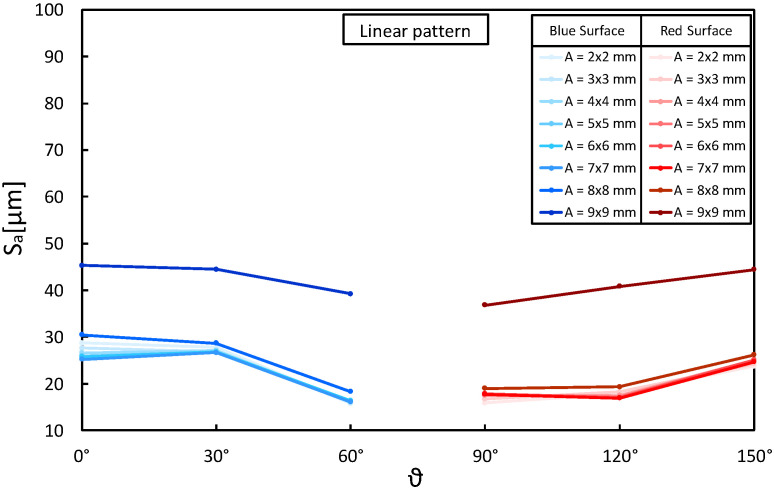
Preliminary measurements on A1, A4, and A7 specimens: evaluation of $S_{a}$ parameter in function of definition area (*A*) and building angle (θ).

**Figure 6 materials-16-00098-f006:**
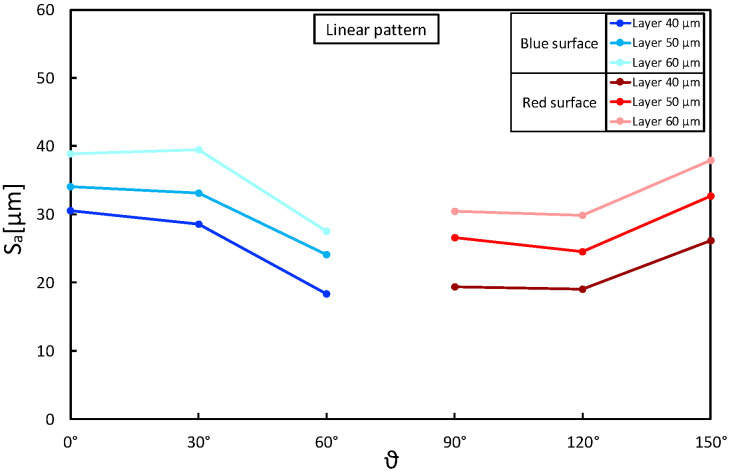
$S_{a}$ as a function of layer thickness (*t*) and building angle (θ): linear laser pattern.

**Figure 7 materials-16-00098-f007:**
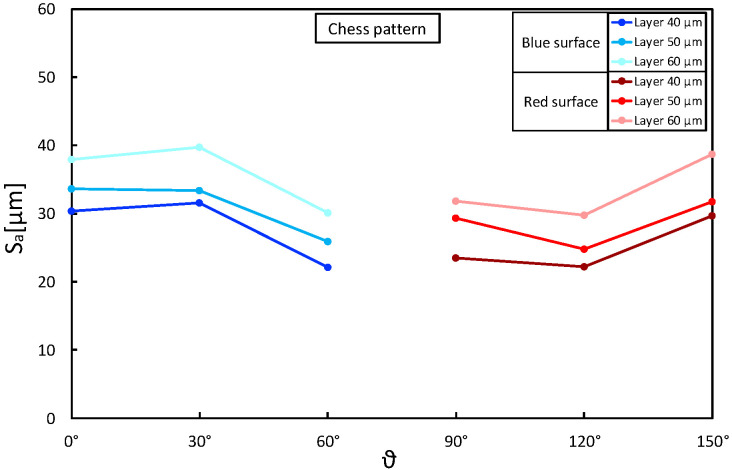
$S_{a}$ as a function of layer thickness (*t*) and building angle (θ): chess laser pattern.

**Figure 8 materials-16-00098-f008:**
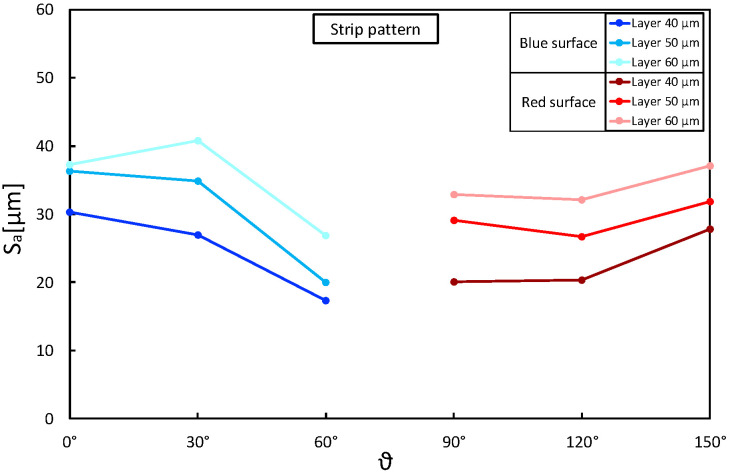
$S_{a}$ as a function of layer thickness (*t*) and building angle (θ): strip laser pattern.

**Figure 9 materials-16-00098-f009:**
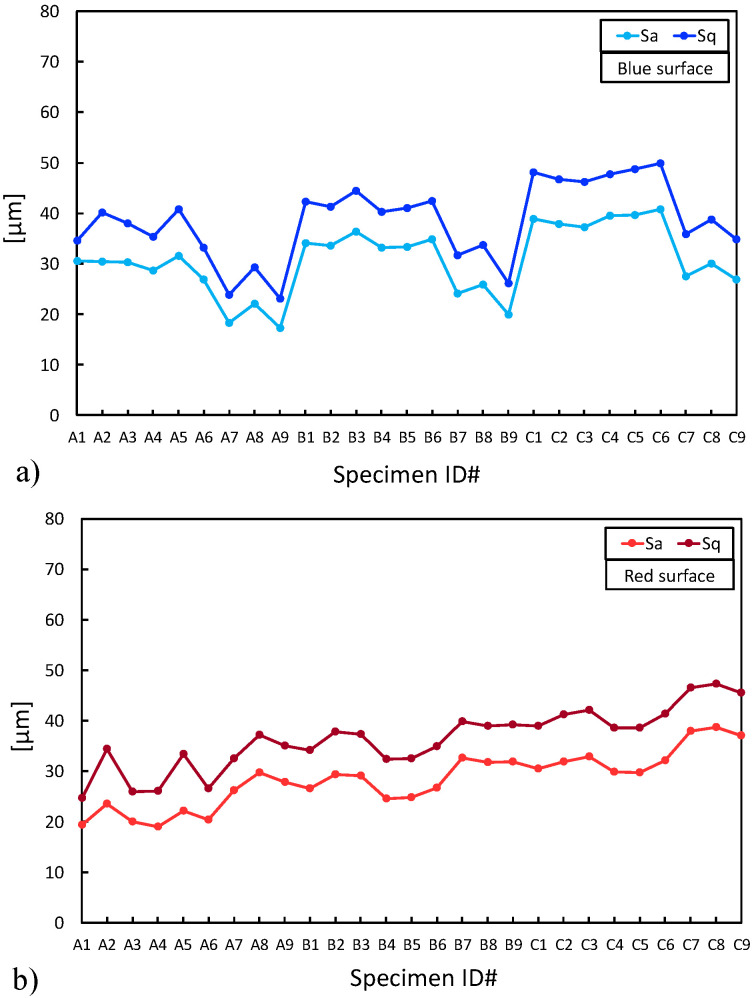
$S_{a}$ and $S_{q}$ comparison: (**a**) blue surface (BS); (**b**) red surface (RS).

**Figure 10 materials-16-00098-f010:**
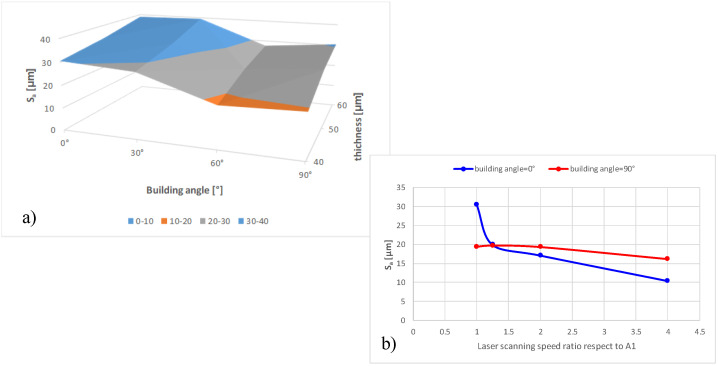
Key results obtained by the proposed experimentation.

**Table 1 materials-16-00098-t001:** Metal4Printing CuCrZr powder chemical analysis wt% (powder size 15–45 µm) [24].

Element	Min		Max
Cr	0.5		1.2
Zr	0.03		0.3
Fe		0.08	
Si		0.01	
Cu		Base	

**Table 2 materials-16-00098-t002:** Full Factorial design of experiment (DOE) of the SLM process parameters.

Print Job	N.	Specimen #ID	Variables & Level (−1, 0, +1)		
			Layer Thickness t [µl	Building Angle [θ]	Laser Pattern
1	1	A1	40	0°/90°	Linear
	2	A2	40	0°/90°	Chess 3 × 3 mm
	3	A3	40	0°/90°	Strip 3 mm
	4	A4	40	30°/120°	Linear
	5	A5	40	30°/120°	Chess 3 × 3 mm
	6	A6	40	30°/120°	Strip 3 mm
	7	A7	40	60°/150°	Linear
	8	A8	40	60°/150°	Chess 3 × 3 mm
	9	A9	40	60°/150°	Strip 3 mm
2	10	A10	50	0°/90°	Linear
	11	A11	50	0°/90°	Chess 3 × 3 mm
	12	A12	50	0°/90°	Strip 3 mm
	13	A13	50	30°/120°	Linear
	14	A14	50	30°/120°	Chess 3 × 3 mm
	15	A15	50	30°/120°	Strip 3 mm
	16	A16	50	60°/150°	Linear
	17	A17	50	60°/150°	Chess 3 × 3 mm
	18	A18	50	60°/150°	Strip 3 mm
3	19	A19	60	0°/90°	Linear
	20	A20	60	0°/90°	Chess 3 × 3 mm
	21	A21	60	0°/90°	Strip 3 mm
	22	A22	60	30°/120°	Linear
	23	A23	60	30°/120°	Chess 3 × 3 mm
	24	A24	60	30°/120°	Strip 3 mm
	25	A25	60	60°/150°	Linear
	26	A26	60	60°/150°	Chess 3 × 3 mm
	27	A27	60	60°/150°	Strip 3 mm

**Table 3 materials-16-00098-t003:** Parameters for the determination of surface texture by areal methods [27].

Parameter	Symbol	Definition
Arithmetical mean height	$S_{a}$	Arithmetic mean of the absolute of the ordinate values
Maximum peak height	$S_{p}$	Largest peak height value
Maximum pit height	$S_{v}$	Minus the smallest pit height value
Maximum height	$S_{z}$	Sum of the maximum peak height value and the maximum pit height value
Skewness	$S_{sk}$	Quotient of the mean cube value of the ordinate values and the cube of $S_{q}$
Kurtosis	$S_{ku}$	Quotient of the mean quartic value of the ordinate values and the fourth power of $S_{q}$

**Table 4 materials-16-00098-t004:** Dimensional control of SLM-manufactured specimens.

Print Job	Specimen	Specimen Dimension		
		x [mm]	y [mm]	z [mm]
I	A1	9.14	9.12	8.94
	A2	9.15	9.14	9.1
	A3	9.05	9.14	9.19
	A4	8.91	9.12	9.12
	A5	9.08	9.11	9.12
	A6	9.11	8.97	9.11
	A7	9.12	9.03	9.11
	A8	9.04	9.11	9.12
	A9	9.07	9.12	9.1
II	A10	9.2	9.18	9.17
	A11	9.17	9.18	9.17
	A12	9.21	9.19	9.2
	A13	9.17	9.1	9.08
	A14	9.16	9.09	8.99
	A15	9.16	9.09	9.17
	A16	9.16	9.04	9.18
	A17	9.17	9.03	9.12
	A18	8.99	9.16	9.17
III	A19	9.11	9.23	9.07
	A20	8.98	9.22	9.09
	A21	9.24	9.23	9.21
	A22	9.21	9.15	9.06
	A23	9.22	8.89	8.85
	A24	9.22	8.89	9.06
	A25	9.21	9.05	9.22
	A26	9.05	9.21	9.23
	A27	9.09	8.98	9.17
	mean	9.13	9.10	9.12
	std. dev.	±0.08	±0.09	±0.09

**Table 5 materials-16-00098-t005:** Assessment of the repeatability of the SLM production process: blue surface (BS).

Specimen #ID	Blue Surface (BS) − Definition Area (*A*) = 8 × 8 mm						
	$S_{a}$ [µm]	$S_{p}$ [µm]	$S_{v}$ [µm]	$S_{z}$ [µm]	$S_{q}$ [µm]	$S_{\mathrm{sk}}$	$S_{\mathrm{ku}}$
A1-1	30.5	195.6	117.3	313.5	34.6	0.33	0.36
A1-2	30.5	194.9	89.9	284.8	37.4	0.55	0.07
A1-3	30.7	180.9	81.7	262.6	37.6	0.36	−0.26
**Mean**	30.6	190.5	96.3	287.0	36.5	0.4	0.1
**Std. Dev.**	0.12	8.29	18.64	25.52	1.68	0.12	0.31

**Table 6 materials-16-00098-t006:** Assessment of the repeatability of the SLM production process: red surface (RS).

Specimen #ID	Red Surface (RS) − Definition Area (*A*) = 8 × 8 mm						
	$S_{a}$ [µm]	$S_{p}$ [µm]	$S_{v}$ [µm]	$S_{z}$ [µm]	$S_{q}$ [µm]	$S_{\mathrm{sk}}$	$S_{\mathrm{ku}}$
A1-1	19.4	146.8	112.3	259.1	24.7	0.22	1.1
A1-2	19.5	157.8	105.8	263.6	25.7	0.66	1.54
A1-3	19.1	196.3	104.2	300.5	25.2	0.65	2.04
**Mean**	19.3	167.0	107.4	274.4	25.2	0.5	1.6
**Std. Dev.**	0.21	25.99	4.29	22.71	0.50	0.25	0.47

**Table 7 materials-16-00098-t007:** Surface texture parameters results: BS with a definition area (*A*) = 8 × 8 mm.

Specimen #ID	Building Angle θ	Blue Surface (BS) − Definition Area (*A*) = 8 × 8 mm						
		$S_{a}$ [µm]	$S_{p}$ [µm]	$S_{v}$ [µm]	$S_{z}$ [µm]	$S_{q}$ [µm]	$S_{\mathrm{sk}}$	$S_{\mathrm{ku}}$
A1	0°	30.5	195.6	117.3	313.5	34.6	0.33	0.36
A2	0°	30.4	245.8	96.9	342.7	40.2	1.26	3.1
A3	0°	30.3	193.7	133.1	326.8	38	0.49	0.45
A4	30°	28.6	225.4	118.1	343.4	35.4	0.63	0.41
A5	30°	31.6	253.7	124.2	377.9	40.8	1.17	2.44
A6	30°	26.9	181.9	94.8	276.7	33.2	0.57	0.12
A7	60°	18.3	122.9	110.1	232.9	23.9	0.64	1.19
A8	60°	22.1	242.2	113.1	355.3	29.3	0.92	2.53
A9	60°	17.3	164.7	84.9	249.6	23.1	0.96	1.98
B1	0°	34.1	219.8	120.5	340.3	42.4	0.2	−0.01
B2	0°	33.6	199.7	151.8	351.5	41.2	0.38	−0.14
B3	0°	36.3	202	111.4	313.4	44.5	0.25	−0.31
B4	30°	33.2	201	121.4	322.5	40.3	0.61	0
B5	30°	33.4	236.8	190.1	427	41.1	0.54	0.33
B6	30°	34.8	230.4	114.4	344.8	42.5	0.6	0.09
B7	60°	24.1	178.5	135.6	314	31.7	0.58	1.08
B8	60°	25.9	174.4	146.4	320.8	33.7	0.45	0.82
B9	60°	19.9	157.7	108.8	266.5	26.2	0.67	1.13
C1	0°	38.9	203.7	147.5	351.2	48.2	0.14	−0.21
C2	0°	37.9	230	132.2	362.2	46.7	0.39	−0.11
C3	0°	37.3	200	140.1	340.1	46.2	0.26	−0.1
C4	30°	39.5	195.6	171.1	366.7	47.7	0.43	−0.32
C5	30°	39.7	225.8	173	398.8	48.7	0.45	−0.04
C6	30°	40.8	255.2	155.6	410.7	49.8	0.49	−0.02
C7	60°	27.6	155.5	154.2	309.7	35.9	0.43	0.74
C8	60°	30.1	231.6	163.4	395	38.8	0.41	0.68
C9	60°	26.8	212.5	139.8	352.3	34.9	0.79	1.14

**Table 8 materials-16-00098-t008:** Surface texture parameters results: RS with a definition area (*A*) = 8 × 8 mm.

Specimen #ID	Building Angle θ	Red Surface (RS) − Definition Area (*A*) = 8 × 8 mm						
		$S_{a}$ [µm]	$S_{p}$ [µm]	$S_{v}$ [µm]	$S_{z}$ [µm]	$S_{q}$ [µm]	$S_{\mathrm{sk}}$	$S_{\mathrm{ku}}$
A1	90°	19.4	146.8	112.3	259.1	24.7	0.22	1.1
A2	90°	23.5	294.6	113.4	408	34.4	1.84	7.76
A3	90°	20	142.7	103.9	246.6	25.9	0.37	0.8
A4	120°	19.0	225.1	92	317.1	26	0.93	2.56
A5	120°	22.2	276.3	138.5	414.8	33.4	2.34	9.45
A6	120°	20.3	128.4	160.1	288.5	26.6	0.19	1.03
A7	150°	26.2	183.1	111.4	294.5	32.5	0.7	0.32
A8	150°	29.7	217.2	108.8	325.9	37.2	0.87	1.06
A9	150°	27.8	174.9	227.3	402.3	35.1	0.5	0.84
B1	90°	26.6	156.1	169.3	325.3	34.2	0.28	0.59
B2	90°	29.4	187.9	149.7	337.6	37.8	0.18	0.61
B3	90°	29.1	229.7	150.6	380.3	37.3	0.08	0.55
B4	120°	24.5	223.9	152.5	376.4	32.4	0.73	1.76
B5	120°	24.8	192.7	136.8	329.5	32.5	0.58	1.31
B6	120°	26.7	214.6	161.3	375.8	34.9	0.22	0.94
B7	150°	32.7	186.2	115.8	302.1	39.8	0.59	−0.18
B8	150°	31.7	163.9	121.8	285.7	38.9	0.53	−0.07
B9	150°	31.9	198.7	135.5	334.2	39.2	0.53	0.06
C1	90°	30.5	172	169.1	341.1	39	0.09	0.39
C2	90°	31.8	205.9	185.7	391.6	41.2	−0.05	0.69
C3	90°	32.9	168.2	176.6	344.8	42.1	−0.08	0.34
C4	120°	29.8	151.1	188.1	339.2	38.5	0.35	0.53
C5	120°	29.9	206.9	183.3	390.1	38.6	0.45	0.79
C6	120°	32.1	206.3	209	415.3	41.4	0.19	0.64
C7	150°	37.9	231.6	160.2	391.9	46.5	0.53	−0.03
C8	150°	38.7	207.2	153.1	360.3	47.3	0.44	−0.21
C9	150°	37.1	206.7	184.4	391	45.5	0.29	−0.07

**Table 9 materials-16-00098-t009:** Optimum conditions and main variables for blue surface (BS): $S_{a}$ mean value for each level of variables (−1, 0, +1).

Variables	Level	$\bar{S_{a}}$ Mean Value	$\bar{S_{a}}$ Difference	
Layer thickness **	a. 40	26.2 *	a.−b.	−4.4
	b. 50	30.6	a.− c.	−9.2 **
	c. 60	35.4	b.−c.	−4.8
Building angle **	a. 0°	34	a.−b.	−0.2
	b. 30°	34.3	a.−c.	10.5
	c. 60°	23.6 *	b.−c.	10.7 **
Laser pattern	a. Linear	30.1*	a.−b.	−1.5
	b. Chess 3 × 3 mm	31.6	a.−c.	0.1
	c. Strip 3 mm	30.2	b.−c.	1.6

* Optimum condition; ** Main variables.

**Table 10 materials-16-00098-t010:** Optimum conditions and main variables for red surface (RS): $S_{a}$ mean value for each level of variables (−1, 0, +1).

Variables	Level	$\bar{S_{a}}$ Mean Value	$\bar{S_{a}}$ Difference	
Layer thickness **	a. 40	23.1*	a.−b.	−5.5
	b. 50	28.6	a.−c.	−10.3 **
	c. 60	33.4	b.−c.	−4.8
Building angle **	a. 90°	27.0	a.−b.	−1.5
	b. 120° *	25.5 *	a.−c.	−5.7
	c. 150°	32.6 *	b.−c.	7.1 **
Laser pattern	a. Linear	27.4 *	a.−b.	−1.7
	b. Chess 3 × 3 mm	29.1	a.−c.	−1.3
	c. Strip 3 mm	28.7	b.−c.	0.4

* Optimum condition; ** Main variables.

**Table 11 materials-16-00098-t011:** Surface texture parameters results increasing the laser scanning speed: blue surface (BS).

Specimen #ID	Laser Scan Speed *	Blue Surface (BS) − Definition Area (*A*) = 8 × 8 mm						
		$S_{a}$ [µm]	$S_{p}$ [µm]	$S_{v}$ [µm]	$S_{z}$ [µm]	$S_{q}$ [µm]	$S_{\mathrm{sk}}$	$S_{\mathrm{ku}}$
A1-4	x4.0	10.4	61.2	101.4	162.6	13.5	−0.66	1.53
A1-5	x2.0	17.1	76.7	85.2	162.0	20.9	−0.05	−0.30
A1-6	x1.25	19.9	89.3	90.4	179.7	24.3	−0.21	−0.43

* With respect to the A1 standard laser scanning speed but with the same energy density (*E*∼156 J/mm${}^{3}$)

**Table 12 materials-16-00098-t012:** Surface texture parameters results increasing the laser scanning speed: red surface (RS).

Specimen #ID	Laser Scan Speed *	Red Surface (RS) − Definition Area (*A*) = 8 × 8 mm						
		$S_{a}$ [µm]	$S_{p}$ [µm]	$S_{v}$ [µm]	$S_{z}$ [µm]	$S_{q}$ [µm]	$S_{\mathrm{sk}}$	$S_{\mathrm{ku}}$
A1-4	x4.0	16.1	101.9	86.3	188.2	20.3	0.05	0.13
A1-5	x2.0	19.3	132.5	107.4	239.9	24.9	0.16	0.61
A1-6	x1.25	19.7	106.3	105.6	211.9	25.6	−0.05	0.65

* With respect to the A1 standard laser scanning speed but with the same energy density (*E*∼156 J/mm${}^{3}$)

## Data Availability

Not applicable.

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
