# Peer review of "Experimental Data Collection of Surface Quality Analysis of CuCrZr Specimens Manufactured with SLM Technology: Analysis of the Effects of Process Parameters"

_materials, 2022, doi:10.3390/ma16010098_

Round 1
Reviewer 1 Report
Dear Authors,
Please improve the paper according to the comments.
1. the Authors write that using the classical approach to surface topography analysis is inappropriate (line 55). However, the research presented here relates to the classical ISO analysis of surfaces fabricated with additive technologies. I strongly recommend multiscale methods for such analysis in future research.
2. I suggest extending the state-of-the-art analysis to include the application of multiscale analysis to the evaluation of additively manufactured surfaces including those using SLM technology
https://doi.org/10.1016/j.measurement.2021.109435
DOI: 10.13140/RG.2.1.3677.5441
3. the surfaces made at an angle of 30 and 150 degrees as well as 60 and 120 degrees are analogous in relation to the building platform. it is necessary to explain the differences between these cases and the validity of such analysis, including statistical evaluation due to different results for individual parameters for particular surfaces
4. why Figure 6 does not refer to analogous angles with respect to building platform
5. explain the differences between methods "a" and "c" in figure 1.
6. How many surface measurements were made for each sample and what is the variation in the obtained parameters for each sample?
7. how many unmeasured points there were
8. the statistical analysis should be improved
9. it would be useful to specify the design guidelines
10. discussion and conclusions have to be improved
Kind regards
Reviewer
Author Response
Our point-by-point response is uploaded here.

Reviewer 2 Report
The authors presented a very good and interesting works. Overall, the paper was well presented and structured. The results can easily be understood. However, there are few element which requires further clarification from the authors;
1. Firstly about the objective as stated in the abstract. How we can relate this objective and the title of the paper.
2. It is better to conclude the findings quantitatively as an ending of the abstract.
3. Is there is any confirmation test conducted? As indicated in the conclusion, the results were validated, but I cannot find in the methodology.
4. In page 12, last paragraph before the conclusion. Can I get any comment on the sentence "Several pre-processing......down-skin).
5. Did conclusion addressed the objective set in the abstract?
The strength is on the literature review, which are mostly updated.

Author Response

(The authors gave the same response as above.)

Round 2
Reviewer 1 Report
Due to the layered nature of model manufacturing, analysis of 30- and 150-degree and 60- and 120-degree seems illegitimate. The resulting differences may be caused by the randomness of the distribution of surface irregularities, so it is crucial to carry out more complex statistical studies on a larger number of samples, including the evaluation of repeatability and reproducibility of the process in order to determine the significance of differences between the surfaces.
Figure 6 should refer to analogous angles relative to the building platorm
The study gives mean values for the measurements. What is the standard deviation of the results
There is many times a problem with optical measurements of additivlly produced surfaces, and the software approximates the surface based on the gathered data by filling in the unmeasured points. How many percentage of points were measured for each surface.
It is advisable to indicate recommendations for the evaluated process parameters in terms of optimal or best surface quality
Reviewer 2 Report
Thank you for the revision. Most of the comments had been addressed by the authors successfully. However, I am not agree on how the conclusion was presented e.g. by stating the figure and most importantly not try to highlight the statement which may answer the objectives.
